# Identification of the nuclear localization signal in the *Saccharomyces cerevisiae* Pif1 DNA helicase

Rosemary S. Lee[1], Carly L. Geronimo[2], Liping Liu[1], Jerzy M. Twarowski[1], Anna Malkova[1]*, Virginia A. Zakian[2]*

1 Department of Biology, University of Iowa, Iowa City, Iowa, United States of America, 2 Department of Molecular Biology, Princeton University, Princeton, New Jersey, United States of America

* anna-malkova@uiowa.edu (AM); vzakian@princeton.edu (VZ)

**Data Availability Statement:** All data are in the manuscript and supporting information files.

**Funding:** AM was funded by R35GM127006 grant from NIGMS; VZ was funded by 1R35GM118279

## Abstract

*Saccharomyces cerevisiae* Pif1 is a multi-functional DNA helicase that plays diverse roles in the maintenance of the nuclear and mitochondrial genomes. Two isoforms of Pif1 are generated from a single open reading frame by the use of alternative translational start sites. The Mitochondrial Targeting Signal (MTS) of Pif1 is located between the two start sites, but a Nuclear Localization Signal (NLS) has not been identified. Here we used sequence and functional analysis to identify an NLS element. A mutant allele of *PIF1* (*pif1-NLSΔ*) that lacks four basic amino acids (781KKRK784) in the carboxyl-terminal domain of the 859 amino acid Pif1 was expressed at wild type levels and retained wild type mitochondrial function. However, *pif1-NLSΔ* cells were defective in four tests for nuclear function: telomere length maintenance, Okazaki fragment processing, break-induced replication (BIR), and binding to nuclear target sites. Fusing the NLS from the simian virus 40 (SV40) T-antigen to the Pif1-NLSΔ protein reduced the nuclear defects of *pif1-NLSΔ* cells. Thus, four basic amino acids near the carboxyl end of Pif1 are required for the vast majority of nuclear Pif1 function. Our study also reveals phenotypic differences between the previously described loss of function *pif1-m2* allele and three other *pif1* mutant alleles generated in this work, which will be useful to study nuclear Pif1 functions.

## Author summary

Investigation of proteins involved in maintaining genomic stability is important for understanding how genetic diseases, such as cancer, arise. Our goal was to determine how the *S. cerevisiae* Pif1 helicase that plays important roles in genome maintenance and is conserved from yeasts to humans is transported into the nucleus. We identified a short sequence located near the carboxyl end of *PIF1* as a Nuclear Localization Signal (NLS). Cells expressing Pif1 lacking this sequence were defective in telomere length maintenance, Okazaki fragment processing and break induced replication but had wild type mitochondrial function. We also generated a new mutant allele that is more defective in Pif1 nuclear functions than existing alleles. These results provide a better understanding of the

from NIGMS. The funders had no role in study design, data collection and analysis, decision to publish, or preparation of the manuscript.

**Competing interests:** The authors have declared that no competing interests exist.

regulation of Pif1 nuclear functions and describe a useful strain for future studies of nuclear Pif1.

## Introduction

*Saccharomyces cerevisiae* Pif1 is a multi-functional DNA helicase that is important for the integrity of both nuclear and mitochondrial (mt) DNA (reviewed in [1,2]). Although its exact role in mitochondria has not been determined, mtDNA is quickly lost in cells lacking this helicase. In addition, *pif1Δ* cells are defective in certain types of recombination between mt genomes [3].

Nuclear Pif1 has multiple functions. It negatively regulates telomerase at telomeres and double-strand breaks [4,5], participates in Okazaki fragment processing [6,7], suppresses genome instability at G-quadruplex motifs [8,9], contributes to fork arrest at the replication fork barrier within ribosomal DNA (rDNA) [10], is critical for Break-Induced Replication (BIR) repair of double strand breaks (DSBs) [11,12], promotes fork progression at tRNA genes [13,14] and centromeres [15], resolves converged replication forks [10,16], and affects the abundance of centromeric RNA [15].

Two isoforms of Pif1 are generated from the same mRNA by the use of alternative translational start sites (Fig 1A). The mitochondrial isoform is produced when translation starts at the first AUG (M1). When translation begins at the second AUG (M2), the nuclear isoform is made [4]. Pif1 localization into mitochondria is directed by a mitochondrial targeting signal (MTS) found between the two translation start sites [4]. Previously, we generated two separation of function alleles of *PIF1* [4]. The *pif1-m1* allele, in which the mitochondrial translation start site is mutated, confers normal nuclear Pif1 functions but fails to maintain mtDNA. Cells expressing *pif1-m2*, in which the nuclear translational start site is mutated, are mitochondrial proficient but deficient in nuclear Pif1 activities. However, the *pif1-m2* allele is not a null for Pif1 nuclear functions; for example, rates of gross chromosomal rearrangements (GCR) are higher in *pif1Δ* cells than in *pif1-m2* cells [8].

Although the mitochondrial targeting sequence is known, the mechanism that targets Pif1 to the nucleus has not been characterized. Transport of proteins into the nucleus can be passive or active. Although even some large proteins can enter the nucleus via passive diffusion [17], the efficient transport of many proteins that are $>\sim60$ kDa is facilitated by their association with carrier proteins called importins [18]. Importins recognize a short amino acid sequence, known as a nuclear localization signal (NLS), which is found within the ORFs of a subset of proteins that are destined for the nucleus. Because Pif1 has a molecular weight of ~98 kDa, it likely contains an NLS.

The goal of this study was to create a *PIF1* allele that is defective in nuclear entry but retains wildtype mitochondrial function. We identified four basic amino acids ($^{781}$KKRK$^{784}$) in the carboxy-terminal portion of Pif1 as the core region of a candidate NLS. Deletion of these four amino acids did not affect mitochondrial function or protein abundance but resulted in defects in four of four tested nuclear functions. Insertion of a heterologous NLS from the simian virus 40 (SV40) T antigen suppressed the nuclear phenotypes of the mutant lacking these four amino acids ($^{781}$KKRK$^{784}$). We propose that this newly identified region of *PIF1* is a functional NLS.

## Results

The *S. cerevisiae* Pif1 is a Super Family I (SF1) DNA helicase. All SF1 helicases contain six motifs within a 400–500 amino acid helicase domain (orange bars marked by Roman numerals; Fig 1A). In addition, the helicase domain of Pif1 family helicases contains a 23 amino acid

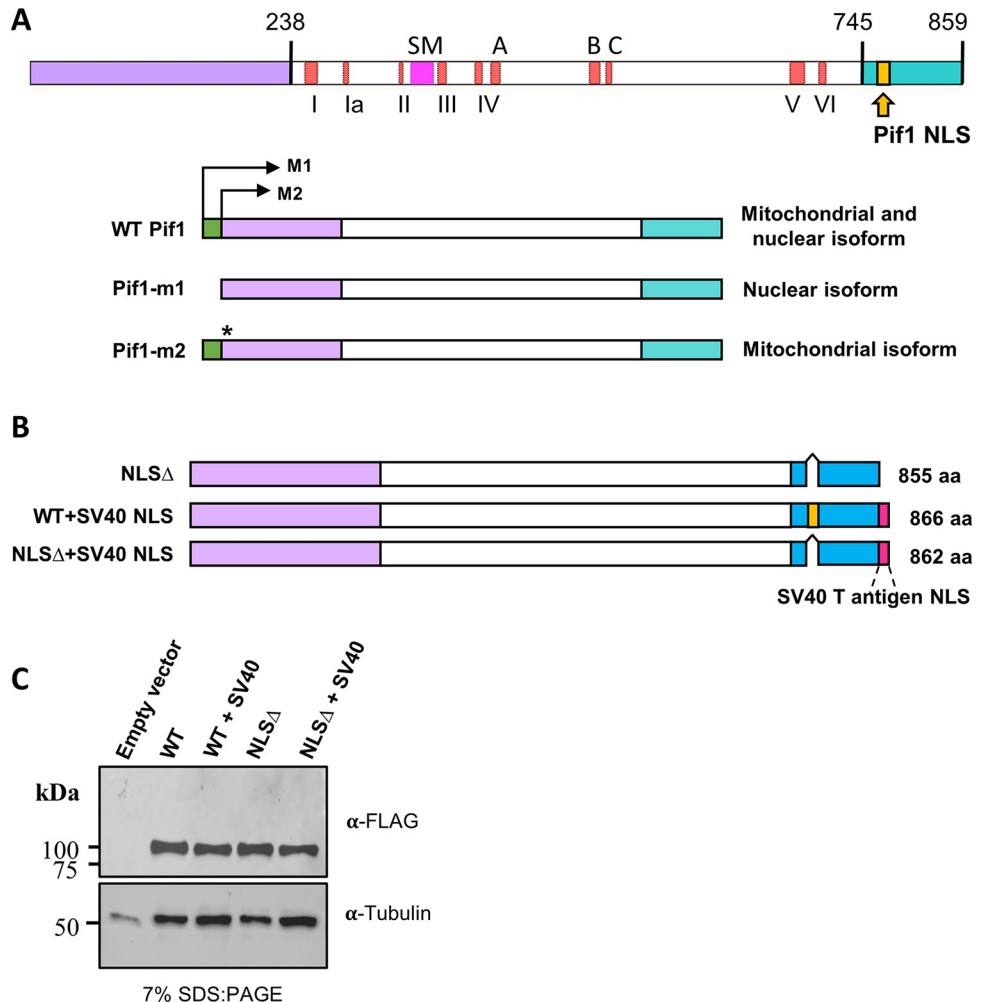

**Fig 1. Structure and expression of wild type and mutant Pif1 proteins.** (A) Structure of wild type Pif1. The 238 amino acid N-terminus is in purple; the 507 amino acid helicase domain is in white and the signature motif (SM) in pink; A, B and C are conserved regions of unknown functions; the 114 amino acid C-terminus is in blue. Numbers above diagram indicate amino acid number. Pif1 family helicases have three domains: an amino-terminal (purple), a helicase core (white), and a carboxyl-terminal (blue) domain. Translation from the first AUG (M1) creates the mitochondrial isoform and translation from the second AUG (M2) creates the nuclear isoform. Located between the two start sites is the mitochondrial targeting signal (MTS, green). A *pif1-m1* allele generated by mutating M1 to alanine creates only the nuclear isoform, while a *pif1-m2* allele generated by mutating M2 to alanine creates the mitochondrial isoform. (B) Structure of mutant proteins. Color scheme is the same as in A. Yellow bars mark the Pif1 NLS while red bar is the NLS from SV40 T-antigen. Lengths of each protein are shown to the right of diagrams. (C) Mutant Pif1 protein expression is comparable to that of wild type Pif1. Westerns analysis of wild type and mutant proteins using an anti-FLAG or anti-alpha tubulin (loading control) antibody on extracts from *pif1Δ* cells carrying an empty vector or the indicated allele of *PIF1*.

signature motif (SM) that is unique to this group of helicases (pink box, Fig 1A) and is essential for ATPase activity [19,20]. In contrast to the helicase domain, the amino (purple rectangle) and carboxyl (blue rectangle, Fig 1A) terminal regions that flank the helicase domain vary greatly in both length and sequence amongst Pif1 family helicases. As part of a strategy to identify separation-of-function *PIF1* alleles, we carried out deletion analysis of the amino and carboxyl terminal regions of the protein. One of the findings from this analysis was that deletion of the terminal 79 amino acids disrupted Pif1 nuclear functions but retained mitochondrial

function [21]. Given that a mutant lacking the same 79 amino acids had ATPase activity *in vitro* [22], we considered that this region might contain the Pif1 NLS.

## Mapping the position of the Pif1 NLS

The cNLS Mapper program was used to search for candidate nuclear localization signals in the Pif1 open reading frame [23]. The program assigns prediction scores of up to 10 with higher scores indicating a higher probability of being an NLS. cNLS Mapper identified two candidate NLSs within Pif1: a monopartite NLS ($^{777}$DEQVKKRKLDY$^{787}$; prediction score of 8) and a bipartite NLS ($^{417}$RQRGDVKFIDMLNRMRLGNIDDETEREFKKLSRP$^{450}$; prediction score of 5). The bipartite NLS had a relatively low prediction score, and its sequence overlapped one of the canonical helicase motifs (motif IV) as well as another conserved region (motif A) (Fig 1A). Based on these findings, we focused our analysis on the monopartite NLS candidate sequence.

## Strains to test whether putative NLS is required for nuclear functions

To determine if the predicted monopartite NLS ($^{777}$DEQVKKRKLDY$^{787}$) was important for Pif1 nuclear functions, we deleted the core of this sequence, $^{781}$KKRK$^{784}$ to generate the *pif1-NLSΔ* (NLS deleted) allele (Fig 1B). Only the core of the putative NLS was deleted to reduce the possibility of disrupting sequences important for other Pif1 functions. In addition, these four amino acids match the consensus for a classical monopartite NLS motif: K-(K/R)-X-(K/R) [24].

As a control, we fused the NLS from the simian virus 40 (SV40) T antigen ($^{126}$PKKKRKV$^{132}$), which functions as an NLS in *S. cerevisiae* [25], to the end of the *pif1-NLSΔ* ORF to generate the *pif1-NLSΔ+SV40* allele (Fig 1B). If our hypothesis is correct, *pif1-NLSΔ* cells should be defective in nuclear Pif1 functions, while still retaining mitochondrial activities. In addition, nuclear defects in *pif1-NLSΔ* cells should be reduced in *pif1-NLSΔ+SV40* cells, even though the SV40 NLS is not in the same position as the predicted Pif1 NLS. We also generated a double mutant of Pif1, *pif1-m2+NLSΔ*, which deleted the candidate NLS from the partial loss of function *pif1-m2* allele.

## Amino acids $^{781}$KKRK$^{784}$ are not essential for protein stability or for maintenance of mitochondrial function

Western blot analysis showed that the Pif1-NLSΔ protein was stably expressed (Fig 1C). In addition, attaching the SV40 NLS to the carboxyl-terminus of either WT Pif1 or Pif1-NLSΔ did not affect the abundance of either protein (Fig 1C).

If $^{781}$KKRK$^{784}$ is the core of a functional NLS, cells expressing *pif1-NLSΔ* should have impaired nuclear functions but normal mitochondrial activity. Yeast requires functioning mitochondria to grow on glycerol media. As expected, *pif1-NLSΔ* and *pif1-NLSΔ+SV40* cells grew as well as WT cells on media containing glycerol as the sole carbon source (Fig 2). This finding indicates that deletion of $^{781}$KKRK$^{784}$ did not affect Pif1 mitochondrial function.

## Amino acids $^{781}$KKRK$^{784}$ of Pif1 are essential for suppressing telomere lengthening

To determine if $^{781}$KKRK$^{784}$ is a functional Pif1 NLS, we examined three nuclear phenotypes: telomere length, BIR, and Okazaki fragment processing in *pif1-NLSΔ* and *pif1-NLSΔ+SV40* cells as well as Pif1 binding to specific sites in nuclear and mitochondrial DNA. Because Pif1 displaces telomerase from telomeres [26], *pif1Δ* cells have long telomeres [4]. Consistent with a

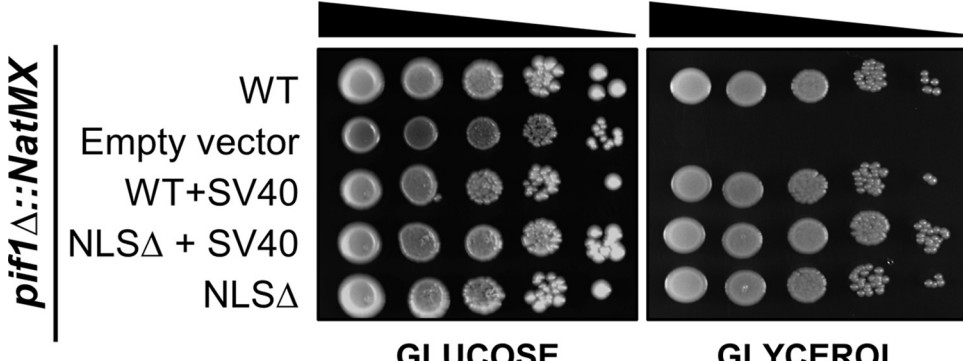

**Fig 2. The Pif1 NLS is not required for its mitochondrial function(s).** Ten-fold serial dilutions of *pif1Δ* cells carrying empty plasmid or plasmids with the indicated *PIF1* allele were spotted on plates containing glucose (left) or glycerol (right). The genomic copy of *PIF1* was replaced with *NatMX*.

lack of nuclear Pif1, *pif1-NLSΔ* cells had long telomeres that were similar in length to those in *pif1Δ* cells (Fig 3). This long telomere phenotype was suppressed in *pif1-NLSΔ+SV40* cells. The ability of the SV40 NLS to suppress the telomere defects of *pif1-NLSΔ* cells supports our hypothesis that [781]KKRK[784] is the core of a functional NLS.

## Pif1's role in break-induced replication (BIR) requires amino acids [781]KKRK[784]

*Pif1* is critical for break-induced replication (BIR) [11,12]. Its proposed roles during BIR include unwinding the DNA duplex to allow the progression of BIR, unwinding the D-loop formed by the newly synthesized DNA, and stabilizing Polδ to enhance BIR processivity [12]. To test the effects of deleting [781]KKRK[784] on BIR [12,27,28], BIR efficiency was determined in derivatives of AM1003 [27], a strain that is well established for BIR assays (Fig 4A). AM1003 and its derivatives are otherwise haploid cells that are disomic for chromosome III and contain a galactose-inducible copy of the HO endonuclease (Fig 4A and S1 Table). One copy of chromosome III, the donor chromosome, is full-length but has a mutant HO recognition site, the *MATα*-inc allele, that is not cleaved by the HO endonuclease. The other copy of chromosome III (recipient) is truncated at the Z-region of *MATa* but contains an intact Ya-region and HO recognition site [27]. When galactose is present and HO is expressed, a DSB is induced at *MATa* on the truncated chromosome III. This DSB is repaired predominantly by BIR that is initiated by strand invasion into the full-length copy of chromosome III (donor). In this system, BIR copying must proceed for ~100 kb to produce a complete BIR outcome, which generates Ade+ Leu- cells (Fig 4A). This system also allows the identification of failed BIR events. In particular, failed DSB repair often leads to the loss of the truncated chromosome and formation of Ade[-red] Leu- colonies (chromosome loss, CL). Another phenotype of failed BIR is Ade[-white] Leu- which is indicative of half-crossover (HC). HCs result from interruption of BIR synthesis, which leads to the fusion between parts of the recipient and donor chromosomes and to the loss of other parts (Fig 4A). In WT (*PIF1*), the majority of DSBs (~ 75%) were repaired by BIR as monitored by the formation of Ade+Leu- colonies, while failed BIR events were less frequent (~9% of CL and ~4% of HC) (Fig 4B and S2 Table).

As shown previously [12,28], the fraction of Ade+Leu- outcomes in *pif1Δ* cells, was significantly reduced (Fig 4B and S2 Table) indicative of defective BIR. Meanwhile, DSB repair classes representing failed BIR (HC and CL), comprised ~40% of all cases, which was significantly higher than in WT cells (p<0.0001).

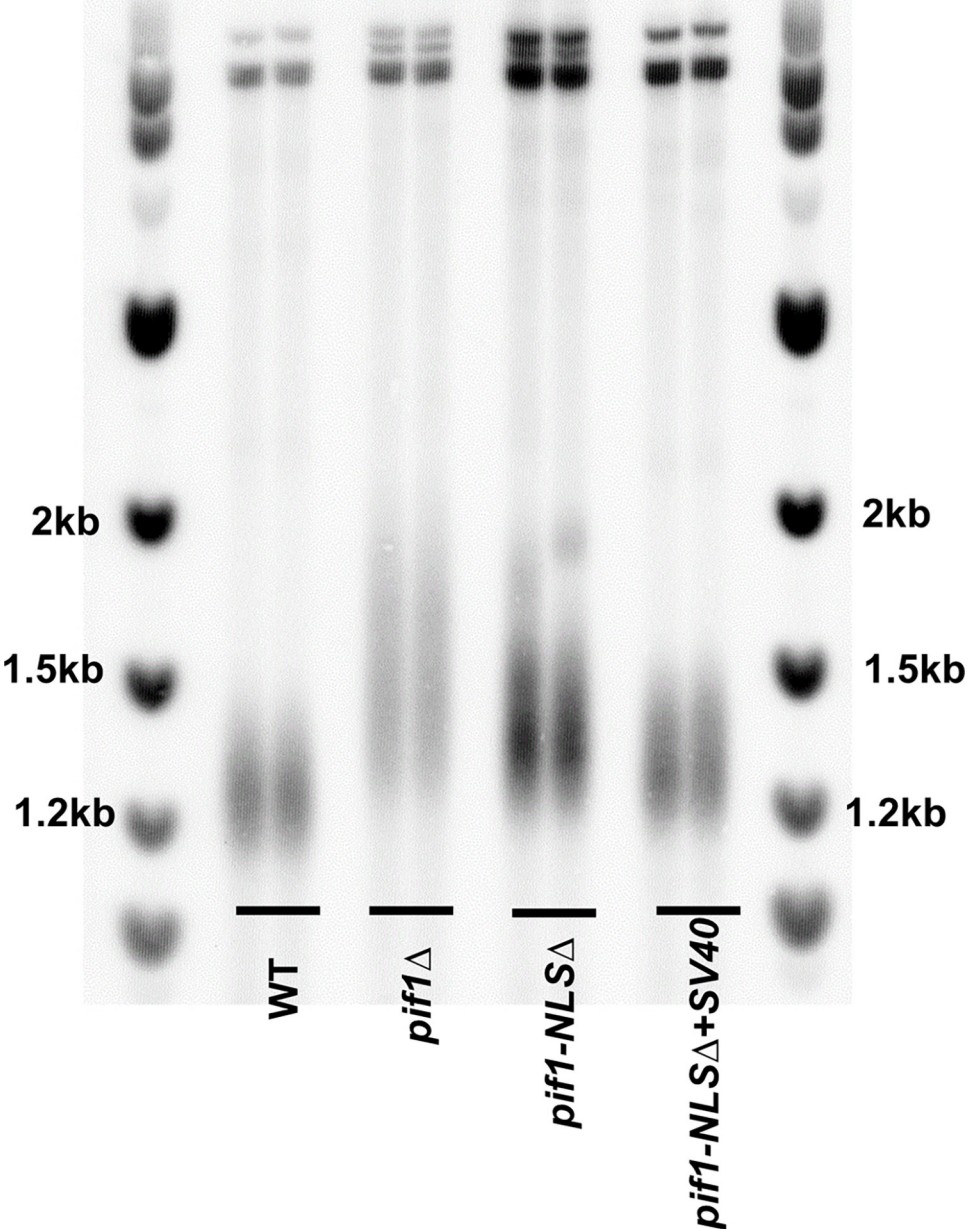

**Fig 3. The Pif1 NLS region is required to maintain wild type telomere length.** Analysis of telomere lengths in two independent isolates of cells expressing *PIF1 (WT)* and various *pif1* mutants. Isolates were streaked at least six times to allow the full impact of each gene on telomere length. Genomic DNA was digested with *XhoI* and analyzed by Southern hybridization using a probe specific to the Y' sub-telomeric region.

Analyses of DSB repair in the *pif1-NLSΔ* mutant showed that the distribution of its DSB repair outcomes was similar to that of *pif1Δ* cells (Fig 4B and S2 Table). In particular, the fraction of Ade⁺Leu⁻ cases was ~45% (significantly lower than in *PIF1* (WT); p < 0.0001), while the fraction of failed BIR events (HC, and CL) was ~45%, significantly higher than in *PIF1* (WT) (p < 0.0001). Importantly, addition of the SV40-NLS largely compensated for the BIR defects in *pif1-NLSΔ* cells (Fig 4B and S2 Table). These data show that [781]KKRK[784] is needed for efficient BIR completion, consistent with the hypothesis that it is the core of a functional

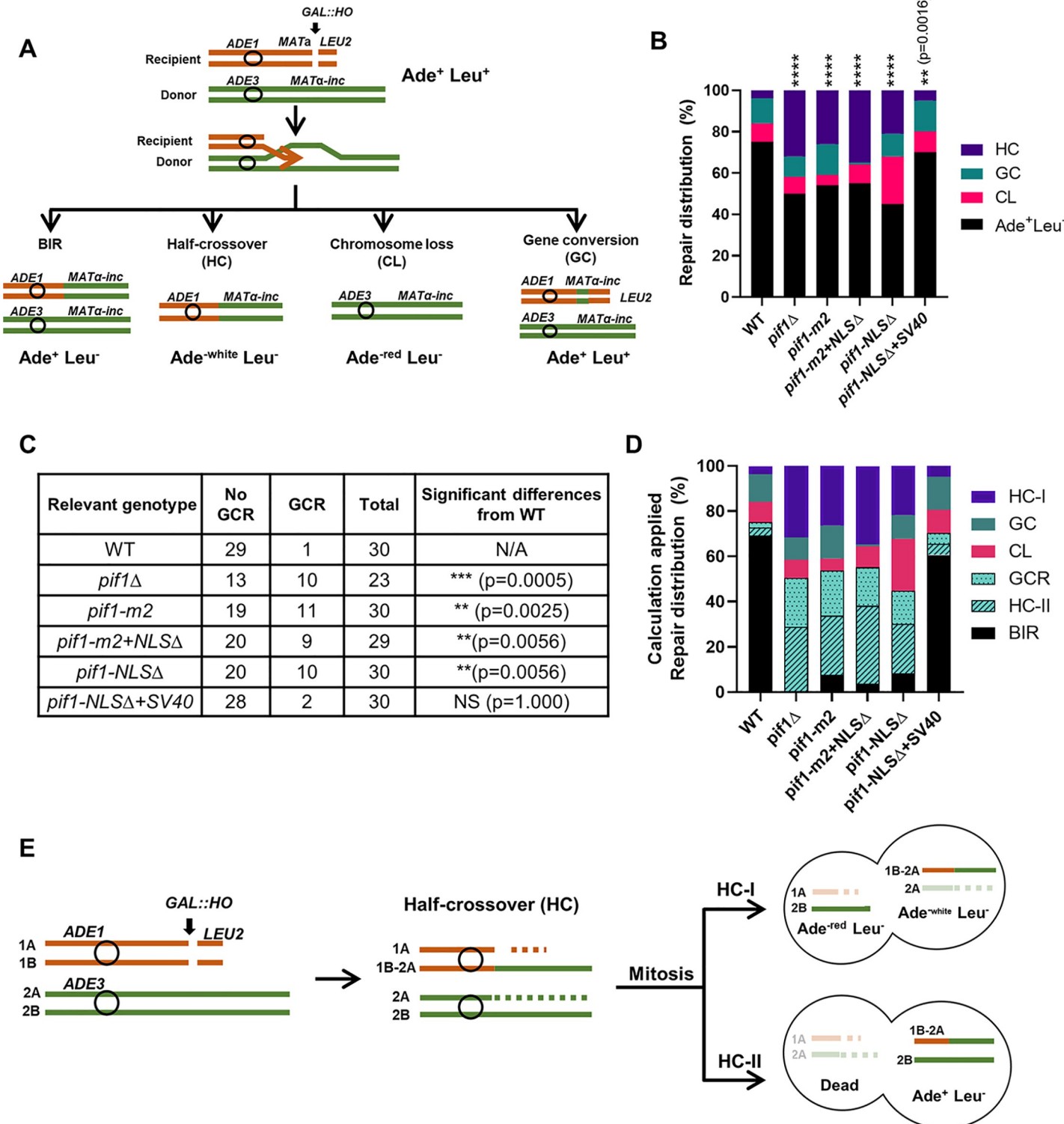

**Fig 4. The effects of *pif1-NLSΔ* on long-range BIR.** (A) Schematic of the BIR genetic assay in a yeast disomic system (AM1003). Repair outcomes are distinguished by genetic markers. (B) The distribution of repair events after DSB induction in WT and indicated mutant strains. Asterisks represent statistically significant difference from WT (*PIF1*) as determined by contingency test (****: p<0.0001; **: p = 0.0016). (C) The fractions of GCR and no-GCR events among Ade⁺Leu⁻ repair outcomes as determined by CHEF gel electrophoresis analysis. Asterisks indicates significant differences from WT and N/A = not applicable; NS = not significant (D) The distribution of all repair DSB events, including calculated % of three classes contributing to the formation of Ade⁺Leu⁻ outcomes (from B), including BIR, half-crossover II (HC-II), and GCRs. The amount of GCRs was calculated by applying the fraction of GCR calculated from C to the total Ade⁺Leu⁻ from B. The number of HC-II was

assumed to be equal to HC-I based on B [27]. (E) An illustration demonstrating the formation of half-crossover events followed by two scenarios of chromatid segregation at mitosis leading to the formation of two half-crossover classes (HC-I and HC-II). In the scenarios shown, only one broken chromatid was repaired while the other was lost. Labels are provided for sister chromatids of both chromosome III homologs (1A, 1B, 2A, and 2B). See Materials and Methods and [27] for details.

NLS. In addition, another separation of function *pif1-m2* mutation and the mutant containing two mutations *pif1*-m2 and *pif1-NLSΔ* were as defective in completion of BIR as *pif1Δ* and *pif1-NLSΔ* (Fig 4B and S2 Table).

## BIR in *pif1-NLSΔ* leads to chromosomal rearrangements

We previously demonstrated that while the majority of Ade$^+$Leu$^-$ colonies in WT cells result from completed BIR events, Ade$^+$Leu$^-$ colonies in *pif1Δ* cells frequently represent gross-chromosomal rearrangements (GCRs) [28]. To determine the fraction of GCRs among Ade$^+$Leu$^-$ colonies, we used contour-clamped homogeneous electric field (CHEF) gel electrophoresis (similar to described in [27]). As expected, we observed a significantly higher fraction of GCR events among Ade$^+$Leu$^-$ colonies in *pif1Δ* cells (10/23, 43%) as compared to WT cells (1/30, 3%) (see Fig 4C for p-values). Likewise, ~36% of Ade$^+$Leu$^-$ events were due to GCR events in *pif1-m2* cells. Similar to *pif1Δ*, ~33% and ~31% of the Ade$^+$Leu$^-$ colonies were due to GCR events in *pif1-NLSΔ* and *pif1-m2+NLSΔ* cells respectively, again, significantly higher than in WT (Fig 4C). In contrast, the frequency of GCR events in *pif1-NLSΔ+SV40* cells was comparable to that of WT cells (Fig 4C).

Moreover, we previously proposed [27] that many of even non-rearranged Ade$^+$Leu$^-$ colonies in *pif1* mutants likely resulted not from completed BIR events, but from the second type of half-crossovers that we call HC-II (see Fig 4D and schematics in Fig 4E for explanation). In brief, HC-II events result from mitotic segregation of half-crossover chromosomes with an intact copy of the donor chromosome, which leads to the formation of Ade$^+$Leu$^-$ colonies. Assuming that the frequencies of Ade$^{-white}$ Leu$^-$ half-crossover (HC-I) and of HC-II classes are equal [27], we can estimate the fraction of HC-II (invisible) class among Ade$^+$Leu$^-$ events for each strain background. Following the subtraction of GCR and HC-II from Ade$^+$Leu$^-$ events, it appears that real BIR events are rarely completed in all *pif1* mutants including *pif1-NLSΔ*, but successfully completed in *pif1-NLSΔ+SV40* (similarly to WT) (Fig 4D).

Together, we conclude that the *pif1-NLSΔ* and *pif1Δ* cells are similarly defective in long range BIR, which requires extensive DNA synthesis (up to 100kb) and which is detected by the BIR assay used here.

## *pif1-NLSΔ* disrupts BIR-associated mutagenesis

To determine if *pif1-NLSΔ* cells are also defective at earlier stages of BIR, we monitored BIR progression within the first 16kb away from the double strand break by measuring the frequency of BIR-associated mutagenesis [28]. For these experiments, we used a *lys2*::$A_4$ reporter gene inserted 16kb centromere-distal to *MATα*-inc. In this system (Fig 5A), Lys$^+$ cells can be generated by frameshift mutations produced within the reporter gene, which are highly stimulated during BIR DNA synthesis [28,29]. Unlike the long-range BIR assay (Fig 4A), a positive signal in the mutagenesis assay (Fig 5A) requires that BIR proceed for only 16 kb.

As in earlier studies [28,29], BIR was associated with a high rate, $3.9 \times 10^{-6}$ of Lys$^+$ events in WT (*PIF1*) cells (Fig 5B and S3 and S4 Tables), and this high rate was galactose dependent (compare 7h versus 0h in S3 and S4 Tables). In contrast, *pif1Δ* cells had a 27x times lower BIR-associated Lys$^+$ mutagenesis than *PIF1* (WT) cells (See S3 and S4 Tables) [29]. Thus, in the absence of Pif1, BIR reaches the 16kb position in only ~4% of the events. Consistent with earlier observations [28], BIR-associated frameshifts at 16kb were 5x more frequent in *pif1-m2*

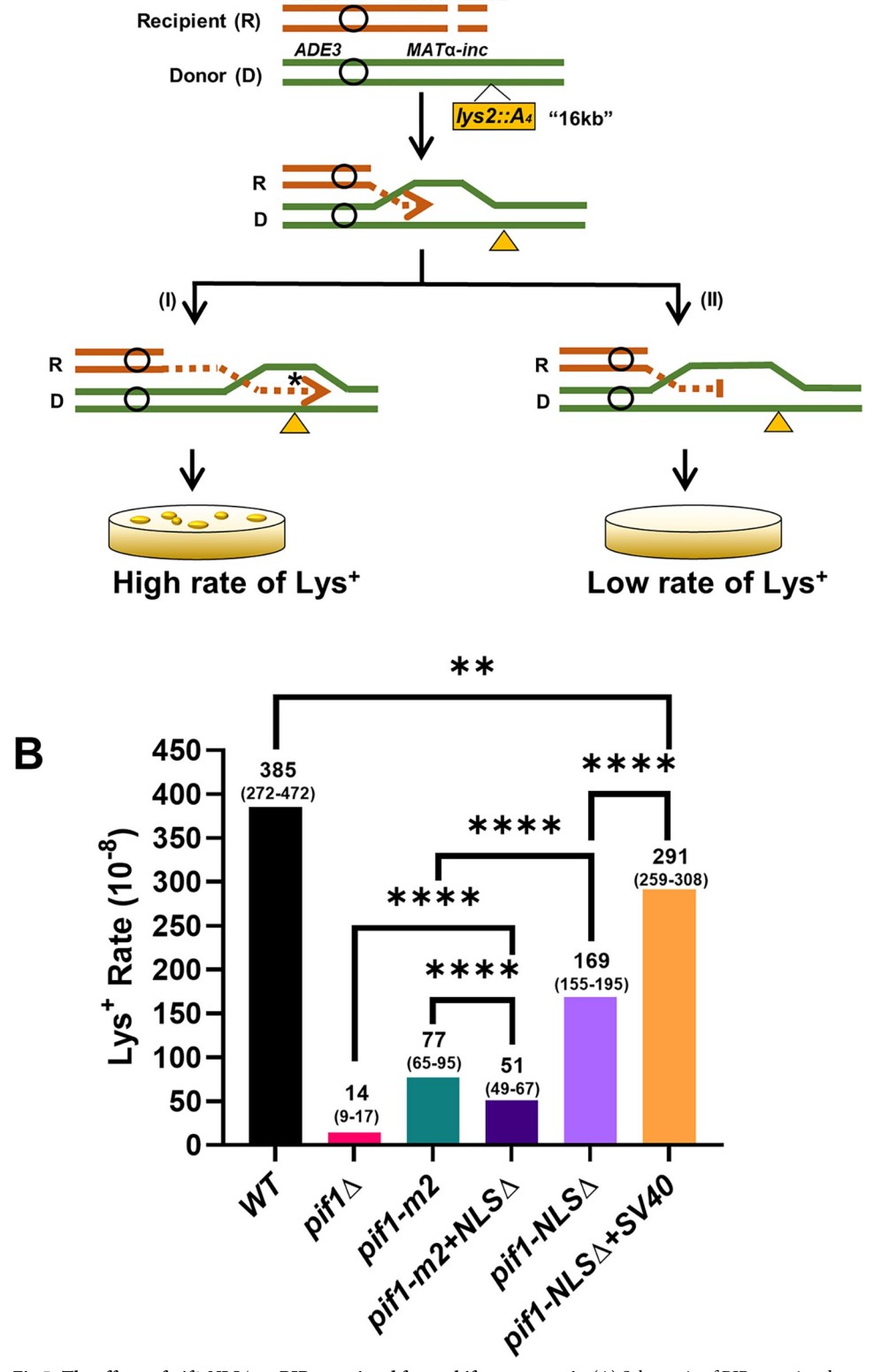

**Fig 5. The effects of *pif1-NLSΔ* on BIR-associated frameshift mutagenesis.** (A) Schematic of BIR-associated mutagenesis assay. The rate of mutagenesis was measured by the frequency of Lys$^-$ to Lys$^+$ reversion using a *lys2::A$_4$* reporter: A -1bp frameshift within the *lys2::A4* reporter generates a Lys$^+$ phenotype. The reporter is inserted 16kb

centromere distal to *MATα*-inc on the donor chromosome. Thus, BIR events that proceed at least 16 kb can generate Lys⁺ cells (I) and results in high rate of Lys⁺. When BIR does not reach 16kb, it cannot generate Lys⁺ cells, and the rate of Lys⁺ is low (II). (B) The rate of Lys⁺ events measured 7hrs after a galactose-induced DSB in cells expressing different *PIF1* alleles. Asterisks (*) indicates statistically significant differences from WT (*PIF1*), determined by the Mann-Whitney test (****: p<0.0001; **: p = 0.0089). See also S3 and S4 Tables for the rates of Lys⁺ reversion prior to DSB and for the details of statistical analysis.

than in *pif1Δ* cells, and this level was 5x lower than in WT (Fig 5B and S3 and S4 Tables), confirming previous data that *pif1-m2* is not a null allele in this assay [8,28,30].

The level of BIR-associated mutagenesis in *pif1-NLSΔ* cells was 2.3x lower than in WT (p<0.0001), and 2.2x higher than in *pif1-m2* cells (p<0.0001) (Fig 5B and S3 and S4 Tables). Thus, like *pif1-m2*, *pif1-NLSΔ* was defective but not a null in the mutagenesis assay. Moreover, *pif1-NLSΔ* cells were not as deficient as *pif1-m2* cells. In addition, the *pif1-m2+NLSΔ* double mutant was significantly more defective as compared to either single mutant (Fig 5B and S3 and S4 Tables). Nonetheless, even this double mutant was 3.6x less defective than *pif1Δ* cells (p< 0.0001 for all comparisons). Thus, there must be residual nuclear Pif1 even in *pif1-m2+NLSΔ* cells. Finally, the level of mutagenesis was restored in *pif1-NLSΔ*+SV40 compared to *pif1-NLSΔ* (p<0.0001), but not to the wild type level (p = 0.0089) (Fig 5B and S3 and S4 Tables).

Together, we conclude that all four *pif1* mutants, including *pif1-NLSΔ*, were highly BIR defective as seen by both long- and short-range BIR assays. However, results from the short-range BIR assay indicated that the three *pif1* mutants are not equally defective for BIR.

## BIR-associated DNA synthesis is defective in *pif1-NLSΔ* cells

Although both the long- and short-range BIR assays revealed that *pif1-NLSΔ* cells are BIR-defective, neither assay measures BIR synthesis directly. Therefore, we used the digital droplet PCR (ddPCR) based AMBER (Assay for monitoring BIR elongation rate) assay, which precisely measures the amount of DNA synthesized during individual BIR events [31,32] by monitoring copy number changes at different BIR positions and at multiple time points after galactose addition (Fig 6A, see legend for details). AMBER uses donor-specific primer sets for three separate locations within a 61 kb BIR region and then normalizes these values to the signals obtained at the *ACT1* locus to determine relative copy number. Primer sets were located at 1.9kb (P2), 22.4kb (P3), and 61.4kb (P4) centromere distal to the DSB. Copy numbers at these positions are 1x before BIR and can increase to a maximum of 2x following BIR. Evidence of DNA synthesis was defined as an increase in donor DNA copy number of at least 1.1x. At 10hrs after DSB induction, WT *PIF1* cells had a 1.7x copy number at P2 and P3 and 1.6x at P4 (See Fig 6B and 6C and S5 Table). In contrast, in *pif1Δ* cells, at 10 hrs, increased copy number was detected only at P2 and was only ~ 1.1x (Fig 6B and 6C and S5 Table), results similar to an earlier report [32]. In *pif1-m2* cells at 10 hrs, a small increase was observed at both P2 and P3 (~1.1x; Fig 6B and 6C), but not at P4. At 10 hrs, *pif1-NLSΔ* cells had increased copy number at all three sites (~1.3x at P2 and ~1.1x at P3 and P4; Fig 6B and 6C). Since DNA synthesis in Pif1-defective mutants is often incomplete and therefore some newly synthesized DNA may degrade by 10 hours, we also compared the maximum copy number increases reached in different strains between 0 and 10 hrs (see Fig 6D) and came to the same conclusions. In addition, as expected, the copy number was higher in *pif1-NLSΔ+SV40* (~1.5x at P2 and ~1.4 at P3 and P4) than in *pif1-NLSΔ* cells. The extent of BIR synthesis was similar in *pif1-m2+NLSΔ* and *pif1Δ* cells (~1.1x at P2; no increase detected at P3 and P4) (Fig 6B, 6C and 6D).

In conclusion, cells lacking ⁷⁸¹KKRK⁷⁸⁴ were defective in BIR initiation and progression and this defective phenotype was largely suppressed in *pif1-NLSΔ+SV40* cells. In agreement

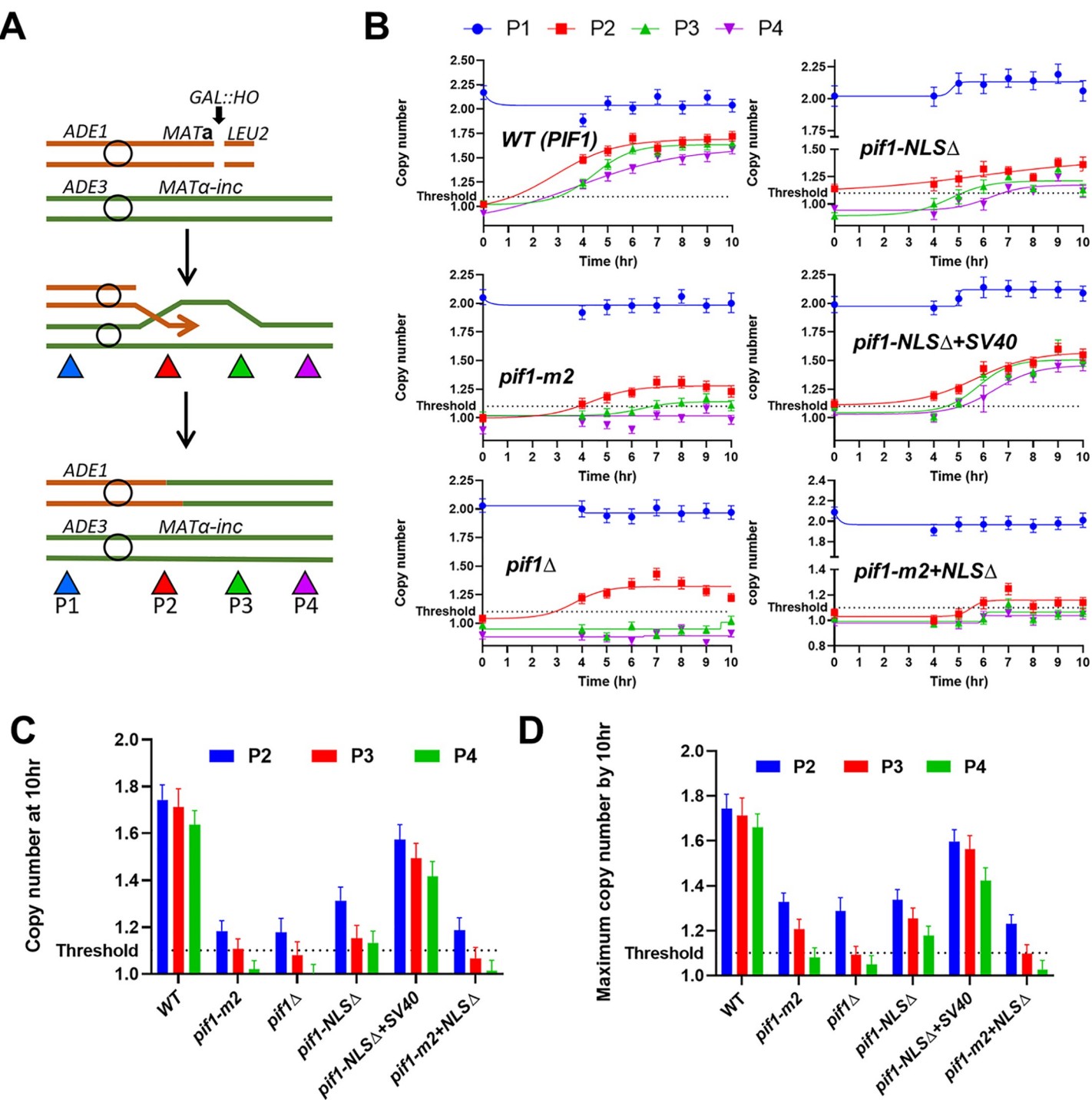

**Fig 6. The effects of *pif1-NLSΔ* on the extent of BIR synthesis detected by AMBER.** (A) Schematic of AMBER analysis measuring the extent of BIR synthesis in a yeast disomic system (derivatives of AM1003). Colored triangles indicate locations of primer sets. (P1 is located at the left arm of chromosome III (~153kb away from *MATα-inc*) and used as a control. P2, P3, and P4 are placed 1.9kb, 22.4kb, and 61.4kb centromere distal from *MATα-inc*. (B) DNA synthesis measured by AMBER in WT (*PIF1*) and in strains containing various mutations of *PIF1* during the first 10hrs following addition of galactose. The results are presented as Boltzmann sigmoidal curve. One out of three independent biological repeats that showed similar results is shown in each panel. (See S5 Table for other repeats). Mean values of target to reference (*ACT1*) loci ratios were calculated by Poisson distribution based on 20,000 droplets with error bars representing upper and lower Poisson 95% CI. (C, D) The comparison between the extents of BIR synthesis in various *pif1* mutants at 10hrs after DSB induction (C) or the maximum detected between 0h and 10hrs (D) as assessed by AMBER. Mean±*s.d* (n = 3, three independent biological repeats). Note: some values are higher in D than in C due to BIR interruption prior to 10h followed by degradation of newly synthesized DNA in *pif1* defective strains.

with the results of BIR-associated mutagenesis assay, these results showed a hierarchy in the BIR defects of the different *pif1* mutants: *pif1Δ ≈ pif1-m2+NLSΔ > pif1-m2> pif1-NLSΔ>pif1-NLSΔ+SV40>*WT.

## $^{781}$KKRK$^{784}$ is important for Pif1 enrichment at nuclear, but not mitochondrial DNA binding sites

The defects in three different nuclear functions suggests that the nuclear abundance of the Pif1-NLSΔ protein was lower than in *PIF1* cells but not as low as in *pif1Δ* cells. We used chromatin immunoprecipitation (ChIP) of Pif1 to its binding sites as a proxy for Pif1 abundance. For these experiments, *PIF1* (WT) and three *pif1* mutant alleles were tagged at their C-termini with 13 MYC epitopes [33]. By western blot, all tagged *PIF1* alleles were stably expressed (Fig 7A).

We analyzed the recruitment of Pif1 during BIR at *MAT* at 4hrs after DSB induction (see Materials and Methods and [12] for details). As expected, we observed a high enrichment of Pif1 at *MAT*, while Pif1-NLSΔ *MAT* binding level was ~4.6x lower than that for WT Pif1

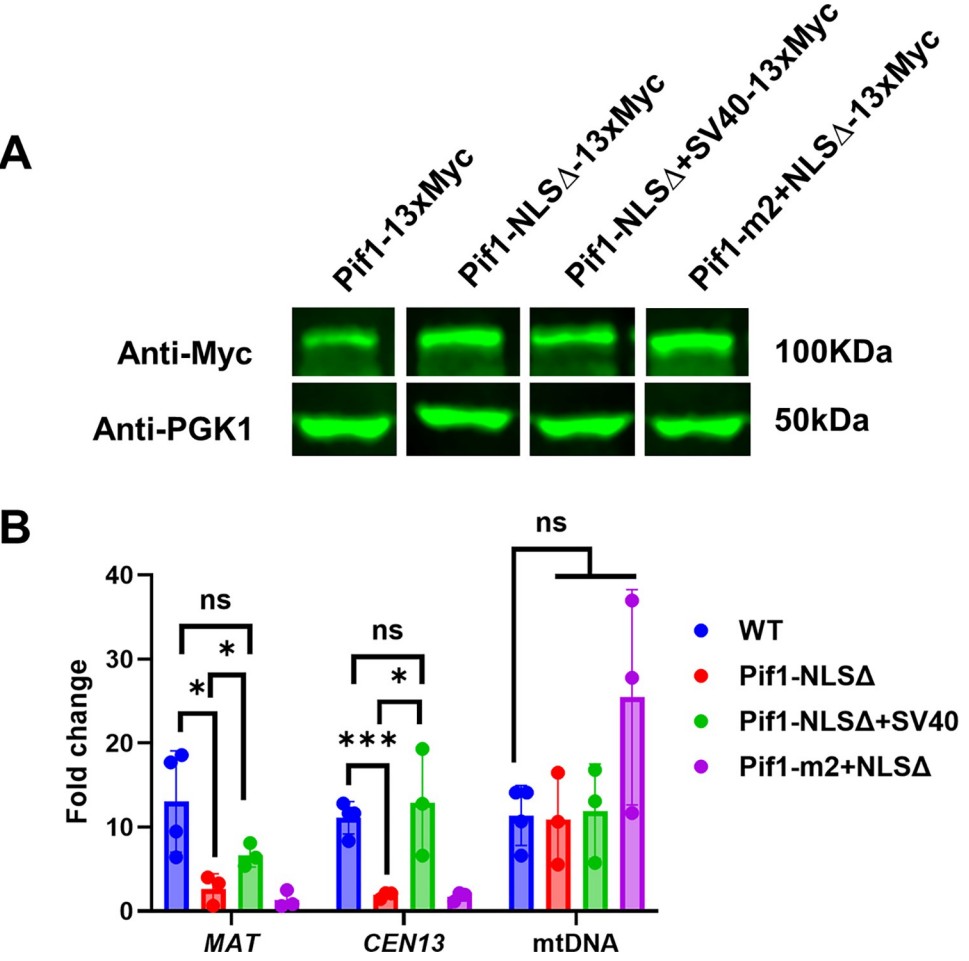

**Fig 7. Diminished enrichment of Pif1 protein lacking $^{781}$KKRK$^{784}$ in the nucleus.** (A) Westerns blot analysis of Myc-tagged *PIF1* (WT) and *pif1* mutant alleles using an anti-MYC or anti-PGK1 (loading control) antibody. (B) Pif1 enrichment measured by ChIP in WT (*PIF1*) and indicated *pif1* mutants at two different nuclear positions *(MAT* (BIR-specific) and *CEN13*) and at mitochondrial-specific site (see Materials and Methods for details). Asterisks represent statistically significant difference as determined by unpaired t-test (***: p = 0.0005, *:p<0.05).

(Fig 7B). We also measured Pif1 binding in cycling cells (no DSB induction) at *CEN13*, a known site of high Pif1 binding [15]. Again, Pif1-NLSΔ binding to *CEN13* was ~4.9x lower than WT Pif1.

As predicted, defective nuclear Pif1-NLSΔ binding was restored in *pif1-NLSΔ+SV40* cells at both *MAT* and *CEN13* sites. As expected, Pif1 binding to a mitochondrial location was indistinguishable in *PIF1* (WT) and three *pif1* mutant cells (*pif1-NLSΔ*, *pif1-m2+ NLSΔ*, and *pif1-NLSΔ+SV40*) (Fig 7B). Together, ChIP data support the hypothesis that *pif1-NLSΔ* leads to a decrease of Pif1 localization to the nucleus but does not affect its presence in mitochondria.

## Partial suppression of *dna2Δ* lethality by *pif1-NLSΔ*

*PIF1* deletion rescues the lethality of *dna2Δ* cells [6]. This rescue is thought to reflect the role of Pif1 in Okazaki fragment maturation. The current model suggests that Pif1 and DNA Polymerase δ together generate long Okazaki flaps that cannot be cleaved by flap endonuclease 1 (Fen1). Rather, the long flaps are cleaved by Dna2, an essential multi-functional helicase-nuclease [34,35]. According to this model, if Pif1 is absent, long flaps are not generated, and Dna2 is no longer essential. For Pif1 to promote formation of long flap Okazaki fragments, Pif1 must be nuclear-localized.

To test whether *pif1-NLSΔ* rescued the lethality of *dna2Δ*, we first used a qualitative assay where colony formation of *pif1Δ dna2Δ* cells transformed with centromeric plasmids containing various *pif1* alleles was measured. As expected, *dna2Δ* cells carrying either the *PIF1*(WT) vector or the vector with *pif1-NLSΔ+SV40* were not viable. In contrast, empty vector alone or vector carrying *pif1-NLSΔ* suppressed the lethality of *dna2Δ* (Fig 8A; compare *NLSΔ* (bottom) to "Empty vector" (top)).

To investigate suppression of *dna2Δ* in a more quantitative assay, we created two diploid strains that were both *DNA2/dna2*::*URA3* heterozygotes. One of these strains was also *PIF1/pif1*::*KANMX*, while the other was *pif1-NLSΔ/pif1*::*KANMX*. Both diploid strains were sporulated and dissected. As expected, in the strain heterozygous for *PIF1* (WT), there were no viable Ura[+] G418[S] (*dna2ΔPIF1*) *s*pores (out of 36 dissected tetrads; Fig 8B). Dissection of 35 tetrads from the *pif1-NLSΔ/pif1*::*KANMX* strain produced three viable Ura[+]G418[S] (*dna2Δpif1-NLSΔ*) spores while 24 Ura[+] spores were G418[R] (*dna2Δ pif1Δ*). This result is indicative of a partial suppression of *dna2Δ* lethality by *pif1-NLSΔ*. As expected, in both strains, an equal number of G418[R] and G418[S] was observed among Ura[-] (*DNA2* wt) colonies (Fig 8B). Our conclusion was further supported by the results of random spore analysis in the same diploids where among 42 Ura[+] clones generated after sporulation of the *pif1-NLSΔ/pif1*::*KANMX* diploid, eight were G418[S] (*dna2Δpif1-NLSΔ*) while 34 were G418[R] (*dna2Δpif1Δ*) (Fig 8C). As expected, no G418[S] Ura[+] (*dna2ΔPIF1*) colonies were observed in the strain heterozygous for *PIF1* (WT) (Fig 8C). Therefore, we conclude that the deletion of *PIF1* [781]KKRK[784] partially suppresses the lethality of *dna2Δ* cells.

## Discussion

The multifunctional Pif1 helicase is critical for maintenance of both nuclear and mitochondrial DNA. Because *pif1Δ* cells grow slowly, *pif1Δ* strains are not ideal for studying nuclear Pif1 functions. Therefore, we sought to identify separation of function alleles that only affect nuclear Pif1. Using cNLS Mapper to identify a putative NLS in Pif1, we generated the *pif1-NLSΔ* allele. As shown here, *pif1-NLSΔ* is a separation of function allele, defective in all tested nuclear functions but having normal localization and function in mitochondria. Together, these data make a strong argument that [781]KKRK[784] is the core of a functional NLS.

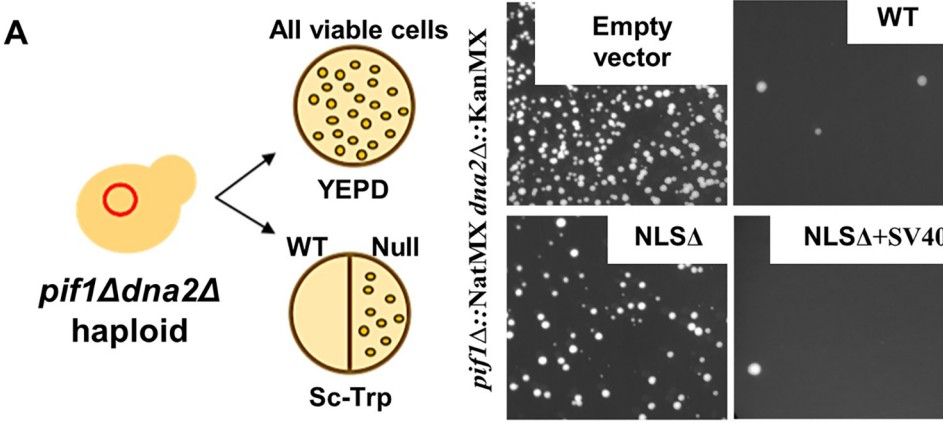

**Fig 8. *pif1-NLSΔ* can partially suppress lethality of *dna2Δ*.** (A) Schematic of experimental system for the qualitative suppression test is shown on the left. Right: representative samples of viability of *pif1Δ dna2Δ* cells carrying empty plasmid or plasmids with the indicated *PIF1* allele. (B, C) The number and phenotypes of viable spores (colonies) obtained following meiosis of two diploids: *DNA2/dna2::URA3 PIF1/pif1::KANMX* and *DNA2/dna2::URA3 pif1-NLSΔ /pif1::KANMX*. The meiotic outcomes were analyzed by tetrad analysis (B) and by random spore analysis (C). G418ˢ = sensitive to G418 (*KANMX* absent); G418ʳ = resistant to G418 (*KANMX* present).

**B**

| Relevant Genotypes | Ura⁺ (*dna2Δ*) | | Ura⁻ (*DNA2*) | |
|---|---|---|---|---|
| | G418ˢ | G418ʳ | G418ˢ | G418ʳ |
| *PIF1/pif1Δ::KANMX* | 0 | 26 | 32 | 38 |
| *pif1-NLSΔ/pif1Δ::KANMX* | 3 | 24 | 34 | 31 |

**C**

| Relevant Genotypes | Ura⁺ (*dna2Δ*) | | Ura⁻ (*DNA2*) | |
|---|---|---|---|---|
| | G418ˢ | G418ʳ | G418ˢ | G418ʳ |
| *PIF1/pif1Δ::KANMX* | 0 | 22 | 43 | 57 |
| *pif1-NLSΔ/pif1Δ::KANMX* | 8 | 34 | 63 | 92 |

The interpretation that *pif1-NLSΔ* cells are defective in nuclear localization of Pif1 is supported by multiple findings. First, cNLS Mapper identified the deleted sequence as having a high probability of being an NLS. Moreover, the core of the predicted NLS, [781]KKRK[784], matches the consensus for a classical monopartite NLS motif: K-(K/R)-X-(K/R) [24]. Second, deletion of [781]KKRK[784] impaired four distinct nuclear functions: telomere length regulation (Fig 3), short- and long-range BIR (Figs 4,5 and 6), binding to nuclear target sites (Fig 7), and Okazaki fragment maturation (Fig 8). Third, all four of the nuclear defects documented in *pif1-NLSΔ* cells were suppressed by addition of the NLS from the SV40 T antigen, which was inserted at a different site within Pif1 than that of the endogenous [781]KKRK[784] motif (Fig 1B). Suppression of defects by a known NLS inserted at a different site in the protein is a classical criterion for identification of an NLS as established by studies in diverse organisms. Fourth, Pif1 localization and activity in mitochondria was not impaired in *pif1-NLSΔ* cells (Figs 2 and 7), as expected if the mutation affects only nuclear localization of Pif1. Fifth, Pif1 binding to nuclear but not mitochondrial binding sites was significantly reduced in *pif1-NLSΔ* cells

(Fig 7). Taken together, these experiments make a strong argument that $^{781}$KKRK$^{784}$ is a functional NLS.

One could still argue that rather than being an NLS, $^{781}$KKRK$^{784}$ might be important for a function other than nuclear localization. Although we cannot completely exclude this possibility, deleting $^{781}$KKRK$^{784}$ does not compromise the known biochemical activity of Pif1. A purified mutant Pif1 lacking its terminal 79 amino acids, including $^{781}$KKRK$^{784}$, has normal ATPase activity *in vitro* [22]. Therefore, the key biochemical activity of the Pif1-NLSΔ protein is almost surely uncompromised in *pif1-NLSΔ* cells. This interpretation is supported by our finding that *pif1-NLSΔ* cells are mitochondrial proficient (Fig 2), while ATPase-dead *pif1-K264A* cells are mitochondrial (as well as nuclear) defective [5].

Another possibility is that $^{781}$KKRK$^{784}$ is required for interaction with a protein(s) required for all four of the nuclear functions assessed in our experiments. Again, we cannot rule out this possibility completely, but we consider it unlikely. First and most critically, we observed that all four nuclear functions were restored by attaching the SV40 NLS, $^{126}$PKKKRKV$^{132}$, to the Pif1-NLSΔ protein. Following earlier studies on NLS characterization, the inserted SV40 NLS was placed at a different position from that of the deleted $^{781}$KKRK$^{784}$ motif (Fig 1). In this new position, the amino acids flanking the SV40 NLS are different from those flanking $^{781}$KKRK$^{784}$ in WT Pif1. Thus, the immediate amino acid context of a hypothetical protein-protein interaction site would be different in WT Pif1 versus Pif1-NLSΔ+SV40. Indeed, the reason we deleted only four amino acids ($^{781}$KKRK$^{784}$) rather than the entire 11 amino acid ($^{777}$DEQVKKRKLDY$^{787}$), computer-predicted NLS, was to minimize undesired changes in the engineered protein. The most parsimonious explanation for suppression of the nuclear defects of *pif1-NLSΔ* cells by insertion of the SV40 NLS at a heterologous site is that the deleted $^{781}$KKRK$^{784}$ is required for the nuclear localization of Pif1. Second, multiple Pif1-interaction experiments, including genetic and biochemical approaches, have been done by us and others (see, for example, [36–38]). To our knowledge, none of these experiments identified a Pif1 interacting protein that is important for all four of the assessed nuclear Pif1 functions. Specifically, it is unlikely that $^{781}$KKRK$^{784}$ is required for Pif1 interaction with PCNA as Pif1 interacts with PCNA via a non-canonical PCNA-binding site that is located distal to and is totally unlike the $^{781}$KKRK$^{784}$ motif [39]. In addition, inhibition of telomerase by Pif1 *in vitro* requires just Pif1 and telomerase [26], while telomerase has no known role in BIR or Okazaki fragment maturation. In addition, this hypothetical Pif1 binding protein would be required for Pif1 binding to its nuclear targets (Fig 7B). We consider it unlikely that $^{781}$KKRK$^{784}$ region is required for interaction with a protein required for all four tested nuclear functions. Therefore, we conclude that the $^{781}$KKRK$^{784}$ motif is unlikely to be a protein-protein interaction site for a protein other than an importin. We still cannot fully exclude that this motif is required for the interaction with another NLS-containing protein, but even if it is true, it would still support our overall conclusion that $^{781}$KKRK$^{784}$ is required for the nuclear import of Pif1.

## The *pif1-NLSΔ* allele is not a null for nuclear functions of Pif1

*pif1-NLSΔ* cells were defective in all tested nuclear functions. In two of these assays, telomere length regulation and long-range BIR, *pif1-NLSΔ* and *pif1Δ* cells were similarly defective (Fig 3 and 4). However, in other more sensitive assays, *pif1-NLSΔ* cells was not a null allele. For example, the very sensitive mutagenesis (Fig 5) and AMBER (Fig 6) assays revealed that both *pif1-NLSΔ* and *pif1-m2* were more active than *pif1Δ*, consistent with earlier findings that *pif1-m2* is not a null allele [8,28,30]. Specifically, the mutagenesis and AMBER assays showed that, in some cells, BIR proceeded for up to 22kb in both *pif1-NLSΔ* and *pif1-m2* cells. In addition, using the AMBER assay, BIR was detected as far as 62 kb from the DSB in *pif1-NLSΔ*

cells, but not in *pif1-m2* cells (Fig 6). Even the double mutant *pif1-m2+NLSΔ* had residual BIR activity in the mutagenesis (but not the AMBER) assay. It is not surprising that the mutagenesis and AMBER assays did not show identical dependencies: the mutagenesis assay detects all completed DSB repair events with BIR synthesis within 16 kb of the DSB, even those that occur with a long delay, as long as they generate viable cells [28]. In contrast, the AMBER assay detects BIR synthesis only during the first 8–10 hours after the DSB [31,32]. On the other hand, the AMBER assay is a direct assay for BIR synthesis and detects not only finished, but also incomplete events [32] while the mutagenesis assay is indirect.

As with the mutagenesis and AMBER assays, *pif1-NLSΔ* cells were not as defective as *pif1Δ* for suppression of *dna2Δ* lethality (Fig 8). Lethality of *dna2Δ* is thought to result from the persistence of long flaps produced by Pif1 during Okazaki fragment maturation. We propose that even a few long flaps resulting from the nuclear localization of small amounts of Pif1-NLSΔ protein is sufficient to kill cells lacking Dna2. Nevertheless, formation of some viable *pif1-NLSΔ dna2Δ* spores suggests that at least some of these cells had no nuclear Pif1-NLSΔ protein.

## Nuclear entry of Pif1 in the absence of the $^{781}$KKRK$^{784}$ motif

Given that residual levels of both BIR and Okazaki fragment maturation occurred in *pif1-NLSΔ* cells (Figs 5,6, and 8), nuclear entry of some Pif1 must occur even in the absence of the $^{781}$KKRK$^{784}$ motif. Likewise, ChIP assays detected Pif1-NLSΔ protein bound to two of two known nuclear target sites, although at both sites, Pif1 bindings was ~5-fold lower than in WT cells (Fig 7). In contrast, Pif1-NLSΔ binding to a mitochondrial site was not reduced (Fig 7).

We envision at least three possibilities for how Pif1 enters the nucleus in *pif1-NLSΔ* cells. The most likely mechanism is that a fraction of the 98kDa Pif1-NLSΔ diffuses into the nucleus, as there is no sharp molecular weight cut off for passive protein import [17]. Another possibility is that small amounts of Pif1 can enter the nucleus via its association with another protein that has its own NLS. Finally, a less active NLS (for example the bipartite NLS identified by cMapper; see Results) may function in a subset of cells. Nonetheless, our data suggest that nuclear entry of Pif1 is heavily reliant on the $^{781}$KKRK$^{784}$ motif. We note that highly sensitive genetic assays are likely better able to detect low levels of nuclear entry compared to cytological or physical assays (such as telomere length). Moreover, small amounts of an enzyme are more likely than small levels of a structural protein to provide detectable function.

## New separation of function *pif1* alleles with impaired nuclear function

A *pif1Δ* strain is not convenient for analysis of nuclear Pif1 functions because of its slow growth. In addition, the defective mitochondrial genomes in different *pif1Δ* strains are not necessarily identical, and impaired mitochondrial function may result in secondary phenotypes that are not due directly to lack of Pif1. Hence, a separation-of-function form of Pif1 that is defective for nuclear, but proficient for mitochondrial functions, would be valuable. As shown here, in two sensitive BIR assays, *pif1-NLSΔ* cells had even more residual BIR activity than *pif1-m2* (Figs 5 and 6). The fact that the doubly mutant *pif1-m2+NLSΔ* allele was more defective than either single mutant in sensitive BIR assays is consistent with the two mutations acting at different steps in Pif1 biogenesis. That is, the *pif1-m2* allele results in reduced translation specifically of the nuclear form of Pif1 [4], while the *NLSΔ* allele reduces nuclear entry of Pif1. Even though there is residual nuclear protein by the most sensitive assays even in *pif1-m2 +NLSΔ* cells, unlike *pif1Δ*, all three of the partial loss of function *pif1* alleles had WT or near WT growth rates and maintained mitochondrial function. Moreover, even though the *pif1-m2 + NLSΔ* allele was not a null in the most sensitive BIR assays, it was more defective than either

single mutation. Thus, a *pif1-m2+ NLSΔ* strain is currently the best one for studying nuclear functions of Pif1.

## Materials and methods

### Yeast strains and growth conditions

To assess the effects of mutant *PIF1* alleles on *dna2Δ* lethality and on mitochondrial proficiency, we used two yeast diploid strains YCG57 or YCG59 (both derived from W303 [40]) that had the following genotypes: YCG57 (*MATa/MATa leu2-3,112/leu2-3,112, trp1-1/trp1-1, can1-100/can1-100, ura3-1/ura3-1, ade2-1/ade2-1, his3-11/his3-11, PIF1/pif1::NatMX6*), YCG59 (*MATa/MATa leu2-3,112/leu2-3,112, trp1-1/trp1-1, can1-100/can1-100, ura3-1/ura3-1, ade2-1/ade2-1, his3-11/his3-11, PIF1/pif1::NatMX6, DNA2/dna2::KanMX6*) [19] (see S1 Table for all genotypes of yeast strains).

The efficiency of BIR and telomere lengths were determined using the AM1003 [27] and its derivatives. These strains are disomic for chromosome III and express the HO endonuclease under the control of a galactose-inducible promoter. The rate of BIR-associated mutagenesis was assessed using derivatives of AM1003 containing a frameshift reporter gene (*lys2::A₄*) inserted into the donor chromosome 16kb centromere-distal to *MATα-*inc *(*AM1291 and its derivatives) [29]. This mutant allele of *LYS2* gene has an insertion of 61bp, leading to a Lys⁻ phenotype due to +1-bp frameshift. A Lys⁺ phenotype is usually restored by a 1bp deletion in the reporter.

The construction of *pif1-m2* and *pif1Δ* derivatives of AM1291 (AM2061 and AM2191, respectively) was described previously [28].

To insert other *pif1* mutant alleles, we first replaced the hygromycin-resistance gene (HPH) located at *HMR* in AM1003, AM1291, and AM2061 with *KANMX* (G418 resistance marker) to create AM5640, AM5978, and AM6121 respectively. To introduce *pif1-NLSΔ*, these strains were co-transformed with an oligo OL5029 (S6 Table) and plasmid pRL10 (see S7 Table; [41]) that contains a region for targeting the NLS region that is complementary to the 50bp 5' of the Pif1 NLS and the 50bp 3' of the NLS region but lacks the NLS itself. Deletion of the *PIF1* NLS region in AM6007 (AM1003 derivative), AM6032 (AM1291 derivative), and AM6201 (AM2061 derivative) was confirmed by PCR and sequencing. To insert the SV40-NLS into the strains containing *pif1-NLSΔ*, the SV40-NLS region was PCR amplified from pCG82 (see S7 Table) using the following primers: OL5032 (S6 Table) homologous to *PIF1* region located centromere proximal to NLS, and OL5033 (S6 Table) where the small letters correspond to the *SV40-NLS* sequence of pCG82 and the capital letters represent homology to the region flanking the *PIF1* gene (located centromere distal from *PIF1*). This PCR product was co-transformed with pRL11 (CRISPR-Cas9 plasmid that contains a region corresponding to 854 aa of Pif1 for CRISPR-Cas9 targeting; see S7 Table). The resulting strains containing the *SV40-NLS* are AM6057 (AM6007 derivative) and AM6060 (AM6032 derivative).

For ChIP analysis, Pif1 alleles were tagged with 13xMyc epitopes that were amplified from the pFA6a-13XMyc-KanMX6 plasmid [33] by using the following primers: RL216 and RL217 (S6 Table). The resulting PCR product was co-transformed with pRL11 into AM6007, AM1003, and AM6201 to generate strains AM6737, AM6740, and AM7068, respectively (S1 Table). In addition, primers RL367 and RL217 were used to amplify SV40 NLS and 13xMyc together, and the PCR product was co-transformed with pRL11 into AM6007 to generate AM7077 (S1 Table).

The constructs were confirmed by Sanger sequencing, and the level of Pif1 in these strains was analyzed by western blot. The construction of diploids used to analyze the suppression of *dna2Δ* lethality by *pif1-NLSΔ* involved the following steps: (i) haploid *MATα-*inc strains were

produced by plating AM6007 and AM1003 on YEP-Gal media followed by selection of Ade[-red] Leu[-] cells to obtain AM6867 (*pif1-NLSΔ*) and AM6870 (*PIF1*) (see S1 Table). (ii) KANMX in AM6867 was replaced by *HPH* to obtain AM6872. (iii) NP265 (*MAT**a** pif1Δdna2Δ*, see S1 Table) was a gift from Dr. Grzegorz Ira lab. AM6872 and AM6870 were crossed to NP265 to generate diploid strains (AM6902 and AM6900, respectively).

## Plasmids

The empty vector used for the analyses of a mitochondrial function and of *dna2Δ* suppression was pRS414 plasmid (*ARS CEN6 TRP1*) [42]. pCG17 is a derivative of pRS414 that contains *PIF1* under control of its endogenic promoter and that was constructed in the following steps: (i) the PspX1/AgeI fragment containing *PIF1* with a carboxy-terminal 3xFLAG tag under control of *RRM3* promoter was cloned into pRS414 digested with PspX1 and AgeI to produce pMB282 [9]; (ii) the *RRM3* promoter in pMB282 was replaced with endogenous promoter of *PIF1* (using digestion with PspX1 and AgeI) to create pCG17 [19]. pCG80 (see S7 Table), which carries the *pif1-NLSΔ* allele, was generated from pCG17 by deleting the sequence corresponding to amino acids [781]KKRK[784] of Pif1. pCG82 was constructed from pCG80 by inserting the region of *SV40* that encodes the SV40 NLS for T-antigen (amino acids [126]PKKKRKV[132]) into the C-terminus of *pif1-NLSΔ*. Plasmids were introduced into yeast cells via lithium acetate transformation [43]. CRISPR-Cas9 plasmids (pRL10 and pRL11), that contain a target region for NLS of Pif1 and carboxy-terminal of Pif1, respectively, were constructed as described [41] and used in construction of *pif1-NLSΔ* (AM6007 and AM6032), *pif1-m2+NLSΔ* (AM6201), and *pif1-NLSΔ+SV40* (AM6057 and AM6060) strains, respectively. pFA6a-13xMyc-KanMX6 plasmid [33] was used as a template to copy 13xMyc to tag various *pif1* alleles for ChIP analysis.

## Media and growth conditions

Depending on the experiment, cells were grown in rich medium YEPD (1% yeast extract, 2% peptone, and 2% glucose) or synthetic drop-out medium lacking tryptophan (Sc-Trp [44]) with glucose (2%) or raffinose (3%). For experiments assessing BIR efficiency or BIR-associated mutagenesis, strains were grown in synthetic drop-out medium lacking leucine with glucose (2%) (Sc-Leu [44]) in rich medium containing lactic acid (3%) instead of glucose (YEP-Lac [45]), followed by addition of galactose directly to the media (to a final concentration of 2%), or plated and on rich medium where glucose was replaced by galactose (2%) (YEP-Gal) [45]. The BIR outcomes were characterized by replica plating on synthetic drop-out media lacking leucine or adenine with glucose (2%) (Sc-Leu and Sc-Ade respectively) [45]. The BIR-associated mutagenesis was assessed by plating on synthetic drop-out medium lacking both adenine and lysine (Sc-Ade, Lys) [45]. To induce meiosis, diploid cells were sporulated by using pre-sporulation medium (YPA, similar to YEPD but potassium acetate was added to 2% instead of glucose) followed replica plating to sporulation medium (SM) that contained 2% potassium acetate [46].

## Identification of the candidate Pif1 NLS motif and construction of Pif1 NLS alleles

The cNLS Mapper (http://nls-mapper.iab.keio.ac.jp/) was used to predict putative nuclear localization signals (NLS) within Pif1 [23]. The core of the functional NLS was narrowed down to a stretch of four basic amino acid residues ([781]KKRK[784]). Deletion of these four amino acid motifs was generated in pCG17 using Quik Change Lightning Site-directed Mutagenesis (Agilent Technologies) to create pCG80. A heterologous NLS ([126]PKKKRKV[132]) from

the SV40 T antigen [25] was introduced into the carboxy-terminal end of pCG17 and pCG80 using gBlocks gene fragments (Integrated DNA Technologies, IDT) and the Gibson Assembly® method (New England Biolabs, NEB). *pif1-NLSΔ alleles* were verified by both restriction enzyme digestion using *Ava*I and *Not*I (NEB) and DNA sequencing (GeneWiz).

## Immunoblot analysis

Yeast total protein extracts were prepared using trichloroacetic acid (TCA) precipitation [47]. Briefly, 10 ml of mid-log phase cells were collected and resuspended in 20% TCA. The cells were lysed with glass beads and vortexing (three 1 min cycles, 4˚C) using the FastPrep-24homogenizer (MP Biomedicals). Two different variations of the method were used after this point:

*Variation A (used for western blot analysis shown in Fig 1C).* The glass beads were washed with 5% TCA and the precipitated proteins were collected by centrifugation. The protein was resuspended in 1X Laemmli buffer (150 mM Tris pH 6.8, 6% SDS, 30% glycerol, 0.3% bromophenol blue, 15% β-mercaptoethanol) and 1M Tris base, then boiled for 5 min. The proteins were resolved on a 7% SDS-PAGE and transferred to nitrocellulose membrane (GE Healthcare). FLAG-tagged Pif1 proteins were detected using mouse anti-FLAG M2 antibody (Sigma-Aldrich) and visualized with HRP-conjugated anti-mouse secondary antibody (Bio-Rad) using standard ECL detection reagents (GE Healthcare) and imaged using the AlphaImager HP system (Protein Simple).

*Variation B (used for western blot analysis shown in Fig 7A).* The precipitated proteins were collected by centrifugation and washed by 0.5M Tris-HCl (pH 8.0). The protein was resuspended in 2x loading buffer (10mM Tris pH 6.8, 4% SDS, 20% glycerol, 0.2% bromphenol blue, 10% β-mercaptoethanol) and 0.5M Tris base (pH 8.0), then boiled for 5 min. The proteins were resolved on 4–20% SDS-PAGE and transferred to nitrocellulose membrane (Amersham #10600002). MYC-tagged Pif1 proteins were detected using antibody (Sigma M4439) and visualized using IRDye 800CW Goat anti-Mouse (#926–32210) from LI-COR. Blots were imaged using an LI-COR Odyssey-Fc Imager.

## Analyzing mitochondrial proficiency

100 ng of plasmid DNA was transformed into a heterozygous *PIF1/pif1Δ*::*NatMX6* diploid strain YCG57 (see S1 Table). After sporulation and dissection, haploid *pif1Δ* spores carrying *TRP1* plasmids were selected and grown to saturation in 5 ml Sc-Trp medium. As previously described [19], serial dilutions were spotted onto Sc-Trp medium with 2% glucose or 3% glycerol and incubated at 30˚C for at least 3 days.

## Telomere length analysis

Genomic DNA was purified from yeast strains as described in [45], and 1.0–1.5ug of this DNA was digested with *Xho*I restriction enzyme (NEB). Following XhoI digestion, DNA was separated by gel electrophoresis in 1.2% agarose gel in TBE buffer, transferred to a nylon membrane (GE Healthcare), and hybridized (as described [45]) to a $^{32}$P-labelled probe specific to Y' pre-telomeric region [48] that was prepared as described in [45]. Blots were imaged using an Azure Sapphire Biomolecular Imager. All strains containing various alleles of *PIF1* were propagated on YEPD plates for at least 100 generations to allow time for establishment of telomere lengths characteristic for each particular *PIF1* allele.

### BIR efficiency assay

Cells were grown in 5ml of liquid Sc-Leu media for approximately 24 hours at 30°C until saturation. Cells were then transferred to YEP-Lac media at ~1-3x10^6 cells/ml and grown for ~ 16 hours until cell density was about ~1-2x10^7 cells/mL. Then cells were plated on YEP-Gal and YEPD plates (at ~50 cells per plate). Plates were incubated for 5–7 days, and then replica plated onto Sc-Ade or Sc-Leu. The frequencies of BIR, gene conversion (GC), half-crossover (HC), and chromosome loss (CL) events were calculated based on respective fractions of Ade$^+$ Leu$^-$, Ade$^+$ Leu$^+$, Ade$^{-white}$ Leu$^-$, Ade$^{-red}$ Leu$^-$ outcomes among all DSB repair events (as described in [45]). Since Ade$^+$ Leu$^-$ can be generated by both BIR and GCRs, the fraction of GCRs among Ade$^+$ Leu$^-$ was estimated by analyzing the representative number of these repair outcomes using CHEF gel electrophoresis (similar to [45]). Due to random segregation of chromosomes, half of the HC products segregate with an intact copy of full-length chromosome III leading to formation of repair products (HC-II) that are genetically indistinguishable from BIR [27]. Therefore, it was assumed that the number of HC-II events is equal to the number of HC-I (Ade$^{-white}$ Leu$^-$ events). Consequently, the number of BIR was adjusted by subtracting the number of HC-II.

### BIR mutagenesis assay

Cells were grown at 30°C in 5ml of Sc-Leu for ~24 hours, diluted 20-fold with YEP-Lac, and then grown for ~16 hours until cell density reached ~1-2x10^7 cells/mL. BIR was induced by addition of 20% galactose to a final concentration of 2%, and cells were incubated for 7 hours after galactose addition. The rate of mutagenesis that occurs during BIR-associated DNA replication was determined by plating cells on Sc-Ade, Lys and Sc-Ade (see [45] for the details of rate calculation). The resulting mutagenesis rate was compared to the rate of mutagenesis that occurs during a normal S phase, calculated by plating cells to the same media prior to galactose addition (see [45] for the details of calculation).

### AMBER (assay for monitoring BIR elongation rate) to detect BIR-associated DNA synthesis

Cells were grown overnight at 30°C in 5ml of Sc-Leu, transferred (1:100 dilution) to YEP-Raffinose and incubated for 16hrs until cell density of ~5x10^6 cells/ml. Then, 50ml of cells were collected (0hr) and stored at -80°C. Galactose was added to the culture to a final concentration of 2% to induce HO endonuclease. During the following 10hrs of incubation at 30°C, ~30ml of cells were collected every hour and stored at -80°C. Genomic DNA was purified and quantified as described for AMBER assay in [31]. For droplet digital PCR (ddPCR) the reaction mix was assembled as follows: 6ul H$_2$O, 1ul of each primer set (one set of primers for *ACT1*, and another–for target position), 10ul of ddPCR supermix (Biorad, #1863026) and 2ul DNA (0.1ng/uL) were added and mixed thoroughly by vortexting. ddPCR was conducted and analyzed as described in [31]. The results were presented as Boltzmann sigmoidal curve. Mean values of target to reference (*ACT1*) loci ratios were calculated by Poisson distribution based on 20,000 droplets with error bars representing upper and lower Poisson 95% CI.

### Assaying the effects of *PIF1* mutants on the viability of *dna2Δ* cells

A qualitative assay for characterization of *dna2Δ* lethality suppression by different alleles of *pif1* has been previously described [19]. In particular 500 ng of plasmid DNA (plasmids included different alleles of *PIF1*) was transformed into a heterozygous *PIF1/pif1Δ::NatMX6 DNA2/dna2Δ::KANMX6* diploid YCG59 (see S1 Table). After sporulation and dissection,

haploid *pif1Δdna2Δ* spores carrying *TRP1* plasmids were selected and grown in 5ml Sc-TRP medium. The cells were plated onto non-selective rich medium YEPD and onto Sc-TRP and grown at 30˚C for 3–4 days. Growth on selective Sc-TRP medium indicates lack of Pif1 nuclear function, as *PIF1* deficiency suppresses the lethality of *dna2Δ* cells [6]. For quantitative assay of lethality suppression, diploid cells that were *DNA2/dna2*::*URA3* and also either *PIF1/pif1*::*KANMX or Pif1-NLSΔ/pif1*::*KANMX* were sporulated and the resulting spores were analyzed by either tetrad analysis (similar to [46]) or by random spore analysis (RSA). For RSA, diploid cells were grown in 5ml of liquid YEPD at 30˚C overnight and transferred to 50ml of liquid YPA media (1:10 dilution ratio) and incubated for ~24hr at 30˚C. Cells were collected and transferred into 50ml of liquid SM media and incubated for at least 72hrs at 30˚C. Cells were spun down and resuspended in 5ml of sterile water. An equal volume of diethyl ether was added to samples, and the mixture was thoroughly mixed for at least 10 minutes at room temperature. Cells were centrifuged and washed by 30ml of sterile water four times, serially diluted, plated on YEPD plates, and incubated for 7days at 30˚C. The phenotypes of meiotic outcomes were classified by replica plating onto Sc-Ade, Sc-Ura, and YEPD+G418 (0.5g/L).

## Chromatin Immunoprecipitation (ChIP) analysis

To perform ChIP assay, 13 repeats of the Myc epitope were inserted at the C-terminus of both *PIF1* and *pif1-NLSΔ*. The levels of Pif1 protein expressed in these two strains was determined by western blot analysis. ChIP analyses was performed as described previously [12] with several modifications. Specifically, yeast cells were grown to a density between $5.0 \times 10^6$ and $1 \times 10^7$ cells/ml in YEP-raffinose and HO-generated DSBs were induced by addition of 20% galactose (to a final concentration of 2%). 40ml of cells were collected before (0h) and after (4h) DSB induction and proteins were crosslinked by the addition of formaldehyde (Sigma #252549) to a final concentration of 1%. Cells were incubated at room temperature for 10 min followed by addition of glycine (125mM final concentration). Cells were incubated for 5 min, and samples were then stored at -80˚C. Cells were lysed with glass beads, and DNA was sheard by sonication to an average size of 0.5kb. Extracts were divided into IP and input samples (9:1 ratio). IP samples were incubated with anti-Myc antibody (Sigma M4439) overnight at 4˚C. Proteins were bound to Dynabeads Protein G (Invitrogen 1004D) and taken through a series of washes and then reverse crosslinked and eluted. Samples were treated with proteinase K followed by phenol DNA extraction and DNA precipitation. Protein enrichment was measured by quantitative-PCR (qPCR) analysis from the input (1:10 dilutions) and IP (undiluted) samples. The following primers were used for qPCR analysis (see S6 Table for details): RL220, RL221 (for *MAT* locus), RL251, RL252 (for *CEN13*), RL304, RL305 (for mtDNA). Both input and IP signals were normalized by using the following primers: RL218, RL219 (*ACT1* locus) used for normalizing data for *MAT*, and RL196, RL197 (*YBL028C* locus) used for normalizing data for *CEN13* and mtDNA.

## Supporting information

**S1 Table. Yeast Strains used in this study.**
(XLSX)

**S2 Table. The effect of *pif1* mutations on the distribution of DSB repair outcomes.** (Related to Fig 4)
(XLSX)

**S3 Table. The rates of BIR-associated mutagenesis in various *pif1*-deficient mutants.** (Related to Fig 5B).
(XLSX)

**S4 Table. The calculation of BIR-associated mutagenesis in various *pif1*-dificient mutants.** (Related to Fig 5B).
(XLSX)

**S5 Table. AMBER analysis of all 3 independent biological repeats.** (Related to Fig 6B)
(XLSX)

**S6 Table. List of oligonucleotides used in this study.**
(XLSX)

**S7 Table. Plasmids used in this study.**
(XLSX)

## Acknowledgments

We are thankful to Dr. Grzegorz Ira for the gift of *dna2Δpif1Δ* yeast strain and for help with ChIP.

## Author Contributions

**Conceptualization:** Rosemary S. Lee, Carly L. Geronimo, Anna Malkova, Virginia A. Zakian.

**Data curation:** Rosemary S. Lee, Carly L. Geronimo, Anna Malkova, Virginia A. Zakian.

**Formal analysis:** Rosemary S. Lee, Carly L. Geronimo, Liping Liu, Anna Malkova, Virginia A. Zakian.

**Funding acquisition:** Anna Malkova, Virginia A. Zakian.

**Investigation:** Rosemary S. Lee, Carly L. Geronimo, Liping Liu, Jerzy M. Twarowski.

**Methodology:** Rosemary S. Lee, Carly L. Geronimo, Liping Liu, Jerzy M. Twarowski.

**Project administration:** Anna Malkova, Virginia A. Zakian.

**Supervision:** Anna Malkova, Virginia A. Zakian.

**Validation:** Rosemary S. Lee, Carly L. Geronimo, Anna Malkova, Virginia A. Zakian.

**Visualization:** Rosemary S. Lee, Carly L. Geronimo.

**Writing – original draft:** Rosemary S. Lee, Anna Malkova, Virginia A. Zakian.

**Writing – review & editing:** Rosemary S. Lee, Anna Malkova, Virginia A. Zakian.

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
