## [Decision Letter · Decision Letter 0]

9 Jun 2023

Dear Anna,

Thank you very much for submitting your Research Article entitled 'Identification of the nuclear localization signal in the Saccharomyces cerevisiae Pif1 DNA helicase' to PLOS Genetics.

The manuscript was fully evaluated at the editorial level and by independent peer reviewers, and I am happy to say that there is consensus among the reviewers that we should proceed to acceptance. However, there is one small data availability issue that still needs to be addressed, regarding the data is Supplemental Table 3, relevant to Figure 5. This table reports rates of Lys+ mutants as median values of a number of repeats. PLOS data availability policy mandates reporting underlying numerical data, which in this case I think are the individual frequencies of Lys+. Please modify supplemental Table 3 so that underneath the summary table (which is excellent, please keep it!), you report the individual values (i.e. Lys+  frequency at 0h and Lys+ frequency at 7h) for each repeat, i.e. for WT, before and after galactose values for repeat 1, repeat 2,...repeat 9. 

Once you have done this:

1) Provide a short note affirming that you have provided this primary data, and describing any other changes you have made in the manuscript.

I regret having you go through the rigamarole of a second submission, but experience has taught us that it is best to make sure that all necessary changes have been made before a manuscript is accepted.

Yours sincerely,

Michael Lichten, Ph.D.

Academic Editor

PLOS Genetics

Gregory Copenhaver

Editor-in-Chief

PLOS Genetics

Reviewer's Responses to Questions

**Comments to the Authors:**

Reviewer #1: The authors have adequately addressed all of my concerns.

Reviewer #2: I am satisfied with the changes in the manuscript.

I have two minor comments.

- To describe the type of statistical method used in Figure 7B and the p-values for * and ***.

- To correct a typo that was found on page 8 on line 145, "carbon course" should be changed to "carbon source".

Reviewer #3: Reviewer 2. Re-review of Lee et al. manuscript

This manuscript is a substantially revised manuscript concerning the identification of a nuclear localization signal in the Pif1 DNA helicase. In my first review of this manuscript, my main two criticisms were that the authors over-stated the strength of their conclusions that the KKRK sequence was the nuclear localization signal and that their description of the effect of mutations of the signal on various aspects of genome stability was confusing.

The basic difficulty (which the authors tried to address) is that the tagged Pif1 protein cannot be visualized in the nucleus. Therefore, a direct argument of the effect of the KKRK motif based on cell biology is not currently possible. However, they make the reasonable argument that the phenotypes associated with the mutant protein are most simply explained by the requirement of the KKRK tag for entry of Pif1 into the nucleus. I agree. I also appreciate their stated caveat in the revised manuscript “We still cannot fully exclude that this motif is required for the interaction with another NLS-containing protein…..”

The revised manuscript also does a better job of discussing the various classes of genetic events induced by a DSB (BIR, chromosome loss, half crossover, GCR, etc.) in their revised Fig. 4. Separating the discussion of the GCR events that occur by BIR in the pif1 mutants from the other events also helps clarify their conclusions.

In summary, the revised manuscript is considerably clearer than the original.

**Have all data underlying the figures and results presented in the manuscript been provided?**

Reviewer #1: Yes

Reviewer #2: Yes

Reviewer #3: Yes

PLOS authors have the option to publish the peer review history of their article (what does this mean?). If published, this will include your full peer review and any attached files.

Reviewer #1: No

Reviewer #2: No

Reviewer #3: No

---

## [Editor Report · Decision Letter 1]

2 Jul 2023

Dear Anna,

We are pleased to inform you that your manuscript entitled "Identification of the nuclear localization signal in the Saccharomyces cerevisiae Pif1 DNA helicase" has been editorially accepted for publication in PLOS Genetics. Congratulations!

Yours sincerely,

Michael Lichten, Ph.D.

Academic Editor

PLOS Genetics

Gregory P. Copenhaver

Editor-in-Chief

PLOS Genetics

Comments from the reviewers (if applicable):

**Data Deposition**

http://datadryad.org/submit?journalID=pgenetics&manu=PGENETICS-D-23-00470R1

**Press Queries**

---

## [Editor Report · Acceptance letter]

20 Jul 2023

PGENETICS-D-23-00470R1 

Identification of the nuclear localization signal in the Saccharomyces cerevisiae Pif1 DNA helicase 

Dear Dr Malkova, 

We are pleased to inform you that your manuscript entitled "Identification of the nuclear localization signal in the Saccharomyces cerevisiae Pif1 DNA helicase" has been formally accepted for publication in PLOS Genetics! Your manuscript is now with our production department and you will be notified of the publication date in due course.

With kind regards,

Zsofia Freund

PLOS Genetics

On behalf of:
